# 2-IPMA Ameliorates PM2.5-Induced Inflammation by Promoting Primary Ciliogenesis in RPE Cells

**DOI:** 10.3390/molecules26175409

**Published:** 2021-09-06

**Authors:** Ji Yeon Choi, Ji-Eun Bae, Joon Bum Kim, Doo Sin Jo, Na Yeon Park, Yong Hwan Kim, Ha Jung Lee, Seong Hyun Kim, So Hyun Kim, Hong Bae Jeon, Hye-Won Na, Hyungjung Choi, Hong-Yeoul Ryu, Zae Young Ryoo, Hyun-Shik Lee, Dong-Hyung Cho

**Affiliations:** 1BK21 FOUR KNU Creative BioResearch Group, School of Life Sciences, Kyungpook National University, Daegu 41566, Korea; muse41191@naver.com (J.Y.C.); kss3213@naver.com (J.B.K.); doosinjo@gmail.com (D.S.J.); yeonie5613@gmail.com (N.Y.P.); yoo035913@gmail.com (Y.H.K.); hajeong1998@naver.com (H.J.L.); kgj010@naver.com (S.H.K.); ks90608@naver.com (S.H.K.); rhr4757@knu.ac.kr (H.-Y.R.); jaewoong64@knu.ac.kr (Z.Y.R.); leeh@knu.ac.kr (H.-S.L.); 2Brain Science and Engineering Institute, Kyungpook National University, Daegu 41566, Korea; loveg730@naver.com; 3Stem Cell Institute, ENCell Co. Ltd., Seoul 06072, Korea; jhb@encellinc.com; 4R&D Center AMOREPACIFIC Corporation, Yongin 17074, Gyeonggi-do, Korea; serina@amorepacific.com (H.-W.N.); heart@amorepacific.com (H.C.)

**Keywords:** primary cilia, 2-IPMA, particulate matter (PM2.5), inflammation, RPE cells

## Abstract

Primary cilia mediate the interactions between cells and external stresses. Thus, dysregulation of primary cilia is implicated in various ciliopathies, e.g., degeneration of the retina caused by dysregulation of the photoreceptor primary cilium. Particulate matter (PM) can cause epithelium injury and endothelial dysfunction by increasing oxidative stress and inflammatory responses. Previously, we showed that PM disrupts the formation of primary cilia in retinal pigment epithelium (RPE) cells. In the present study, we identified 2-isopropylmalic acid (2-IPMA) as a novel inducer of primary ciliogenesis from a metabolite library screening. Both ciliated cells and primary cilium length were increased in 2-IPMA-treated RPE cells. Notably, 2-IPMA strongly promoted primary ciliogenesis and restored PM2.5-induced dysgenesis of primary cilia in RPE cells. Both excessive reactive oxygen species (ROS) generation and activation of a stress kinase, JNK, by PM2.5 were reduced by 2-IPMA. Moreover, 2-IPMA inhibited proinflammatory cytokine production, i.e., IL-6 and TNF-α, induced by PM2.5 in RPE cells. Taken together, our data suggest that 2-IPMA ameliorates PM2.5-induced inflammation by promoting primary ciliogenesis in RPE cells.

## 1. Introduction

Within the homeostasis maintenance process, cellular responses to environmental signals involve multiple and complex communication pathways. Atmospheric pollutants cause serious health problems; indeed, the premature death of 3.7 million people annually worldwide is linked to air pollution [1,2]. Fine particulate matter (PM) in air pollution comprises coarse and fine fractions with aerodynamic diameters < 2.5 μm (PM2.5); these are composed of several molecules including toxic heavy metals, ionic elements, and polycyclic aromatic hydrocarbons. PM can cause epithelium injury and endothelial dysfunction. It can penetrate the nasal cavity and bronchial cilia and thereby induce inflammation, asthma, chronic bronchitis, and renal injury through increased oxidative stress and inflammatory responses [3,4,5]. Moreover, PM can induce eye injury as well as increase the risk of vascular damage and neurotoxicity [6,7].

Primary cilia, i.e., microtubule-based organelles, function as antennae for sensing the extracellular environment and they mediate the interactions between cells and external stimuli including chemical and mechanical stresses [8,9]. Specifically, primary cilia are dynamically regulated, highly conserved organelles that emanate from the surface of most human cells [10]. Their major role is to recognize extracellular signals such as growth factors, nutrients, and hormones; thus, the cilium membrane harbors various receptors, ion channels, and signaling components such as Sonic hedgehog (Shh) and PDGF receptors [9]. Consistently, primary cilia play important roles in signal transduction during development, cell migration, and cell death as well as in the cell cycle. Thus, dysregulation of primary cilia by loss of ciliary proteins increases cell death [11,12]. In contrast, enhanced primary ciliogenesis reduces ischemic injury [13]. Therefore, dysregulation of primary cilia underlies a number of human diseases and syndromic disorders termed “ciliopathies” [9]. For example, degeneration of the retina is a frequently observed clinical ciliopathy caused by dysregulation of photoreceptor primary cilia [14]. As both ADP-ribosylation factor-like protein 13B (ARL13B) and Smoothened (Smo) are localized to primary cilia, these proteins are commonly used as markers to monitor primary ciliogenesis [15]. In addition, intraflagellar transport protein 88 (IFT88), a core anterograde protein, is critical for ciliary assembly and maintenance [16].

Recently, our group showed that PM2.5 reduces the number of primary cilia by increasing cellular stress [17]. Although several regulators involved in primary ciliogenesis have been identified, the role played by primary cilia in PM2.5-mediated cellular stress in retinal pigment epithelium (RPE) cells and the regulators of this process remain poorly understood. In the present study, we identified 2-isopropylmalic acid (2-IPMA) from metabolite library screening as a novel inducer of primary ciliogenesis. In our test, 2-IPMA strongly promoted primary ciliogenesis in RPE cells and restored PM2.5-induced dysgenesis of primary cilia. In addition, we found that 2-IPMA efficiently inhibited oxidative stress and inflammation caused by PM2.5 in RPE cells.

## 2. Materials and Methods

### 2.1. Cell Culture

Human telomerase-immortalized RPE cells and RPE/Smo-GFP cells stably expressing Smo-GFP proteins were kindly provided by Dr. Kim, J (KAIST, Daejeon, Korea). RPE and RPE/Smo-GFP cells were cultured in Dulbecco’s modified Eagle’s medium supplemented with 10% fetal bovine serum and 1% penicillin–streptomycin (Invitrogen, Carlsbad, CA, USA).

### 2.2. Image-Based Fecal Metabolite Library Screening

The RPE/Smo-GFP (3 × 10^3^) cells were seeded in 96-culture-well plates for the image-based fecal metabolite screening library (MetaSci, Toronto, ON, Canada). Following 24 h incubation after seeding, each metabolite of the fecal metabolite library (20 or 100 μM) was added to each well. Cells were then cultured for a further 24 h before cells with activated ciliogenesis were observed under a fluorescence microscope (IX71, Olympus, Tokyo, Japan). SAG was used as a positive control. The experiments were repeated twice with consistent results.

### 2.3. Reagents

PM2.5 was collected on a Teflon filter (Zefluor; Pall Life Science, Ann Arbor, MI, USA) using a low-volume air sampler consisting of a cyclone (2.5-μm size cut, URG-2000-30EH), two upstream denuders (annular, URG-2000-30 × 242-3CSS), a Teflon filter (Zefluor^TM^ 2.0 μm; Pall Life Sciences, Mexico City, Mexico), a backup filter, and a backup denuder in series. The collection started at approximately 10:00 a.m.; the filters were replaced every 24 h and the flow rate was 16.7 L/min. The sampling region was located 35 km southeast of downtown Seoul, Korea (37.34° N, 127.27° E; 167 m above sea level). The filter was sonicated in ethanol (EtOH) for 30 min, the EtOH was evaporated, and then the PM2.5 was resuspended in deionized water. The PM was collected using a high-capacity air collector with a quartz filter. Chemical reagents, including Hoechst 33342, SP600125, 2-IPMA, Anisomycin, and NAC were purchased from Sigma-Aldrich (St. Louis, MO, USA) and SAG (Smoothened agonist) was purchased from Calbiochem (San Diego, CA, USA). Cell Counting Kit-8 (CCK-8) was purchased from Dojindo (Rockville, MD, USA).

### 2.4. Gene Knockdown

For gene expression knockdown, cells were transfected with previously validated siRNA targeting human *IFT88* (5′-CCGAAGCACUUAACACUUA-3′) and negative scrambled siRNA (5′-CCUACGCCACCAAUUUCGU-3′) using Lipofectamine 2000 (#11668019, Thermo Fisher Scientific, Waltham, MA, USA) in accordance with the manufacturer’s protocol. The siRNAs were synthesized from Genolution (Seoul, Korea). At 48 h post transfection, the cells were further treated with the indicated reagents.

### 2.5. Oxidative Stress Measurement

Intracellular ROS levels were measured using a fluorescent dye, 2′,7′-dichlorofluorescein diacetate (H_2_DCF-DA) (Invitrogen, Carlsbad, CA, USA), which is converted into the highly fluorescent 2′,7′-dichlorofluorescein (DCF) in the presence of oxidants. Briefly, cells were plated in 24-well plates. With or without prior application of drugs for 24 h, the cells were then incubated with H_2_DCF-DA (20 μM) in serum-free medium for 40 min (IX71 with 488 nm excitation filter). Relative ROS ratio was presented as the change in fluorescence of drug-treated samples compared with that of control samples.

### 2.6. Western Blot Analysis

All lysates were prepared in 2 × Laemmli sample buffer (62.5 mM Tris-HCl, pH 6.8, 25% (*v*/*v*) glycerol, 2% (*w*/*v*) sodium dodecyl sulfate (SDS), 5% (*v*/*v*) β-mercaptoethanol, and 0.01% (*w*/*v*) bromophenol blue (Bio-Rad, Hercules, CA, USA)). All cellular proteins were quantified using Bradford solution (Bio-Rad) according to the manufacturer’s instructions. The samples were then separated using SDS-polyacrylamide gel electrophoresis and transferred onto a polyvinylidene fluoride membrane (Bio-Rad). After blocking with 4% (*w*/*v*) skim milk in Tris-buffered saline plus Tween (TBST; 25-mM Tris, 140-mM sodium chloride, and 0.05% (*v*/*v*) Tween^®^ 20), the membranes were incubated overnight with the following specific primary antibodies: Actin (Millipore, MA, USA), phosphor-JNK and JNK (Cell Signaling Technology, Danvers, MA, USA), and IFT88 (Proteintech, Chicago, IL, USA). For protein detection, the membranes were incubated with horseradish peroxidase (HRP) conjugated secondary antibodies (Cell Signaling Technology).

### 2.7. Cilia Staining and Counting

For the staining of primary cilia, cells were washed with cold phosphate-buffered saline (PBS) and then fixed with 4% (*w*/*v*) paraformaldehyde dissolved in PBS containing 0.1% (*v*/*v*) Triton X-100. Subsequently, the cells were blocked with PBS containing 1% bovine serum albumin and incubated overnight at 4 °C with primary antibodies against ARL13B (17711-1-AP, 1:1000 Proteintech, Chicago, IL, USA) in 1% bovine serum albumin. After washing, the cells were incubated with Alexa Fluor 488-conjugated secondary antibodies at room temperature for 1 h. The cells were then treated with Hoechst 33,342 dye (H3570, 1:10,000 Thermo-Fisher, Waltham, MA, USA) for nuclear staining. Cilia images were observed using a fluorescence microscope. Cilia were counted in about 200 cells under each experimental condition (*n* = 3). The ciliated cell percentage was calculated as follows: (Total number of cilia/Total number of nuclei in each image) × 100. Cilia lengths were measured using the Free-hand Line Selection Tool of Cell Sense Standards software (Olympus Europa Holding GmbH, Hamburg, Germany) and the average cilium lengths were calculated. Analysis of graph data was performed with GraphPad Prism 8 (GraphPad Software, San Diego, CA, USA).

### 2.8. ELISA Assay

The cytokine levels of cell culture supernatants were determined using ELISA. Human IL-6 and TNF-α in cell culture supernatants were separately determined by ELISA kits from BD biosciences (San Jose, CA, USA) and Abcam (Cambridge, MA, USA) according to the manufacturer’s instructions.

### 2.9. Statistical Analysis

Data were obtained from at least three independent experiments and are presented as means ± standard error of the mean (SEM). Statistical evaluation of the results was performed using one-way analysis of variance. Data were considered significant where *p* values were <0.05.

## 3. Results and Discussion

### 3.1. 2-IPMA Promotes Primary Ciliogenesis in RPE Cells

Primary cilia are strongly associated with various cellular processes such as cell signaling and development [18]. Visual detection systems for biological proteins and metabolites are essential, and Smo protein accumulates on primary cilia; therefore, it is widely used as a cilium marker [10,19]. To identify metabolites for regular ciliogenesis, we previously developed a cell-based screening system using RPE cells that stably express a fluorescent protein fused with Smo (RPE/Smo-GFP) [20]. Using this screening system, we screened a library containing ~550 fecal metabolites. From the screening, we identified 2-isopropylmalic acid (2-IPMA) as well as several other putative candidates. IPMAs are intermediate in the biosynthesis of leucine (Figure 1A) [21]; however, the effects of IPMAs on primary cilia have not yet been elucidated. To verify the screening results, RPE/Smo-GFP cells were treated with either SAG or 2-IPMA at different concentrations (20 and 100 µM) (Figure 1B). SAG, an agonist for Smoothened, was used as a positive control to increase primary ciliogenesis. As shown in Figure 1B,C, treatment with 2-IPMA strongly increased the formation of primary cilia in RPE/Smo-GFP cells (Figure 1B,C). Moreover, the increased primary ciliogenesis was efficiently blocked by depletion of IFT88 expression (Figure 1D,E), suggesting that 2-IPMA is a strong inducer for primary ciliogenesis. In addition, a cell viability assay showed that 2-IPMA is not cytotoxic at a high concentration (~500 µM) in RPE cells (Figure 1F).

### 3.2. 2-IPMA Inhibits PM2.5-Mediated Ciliary Dysgenesis in RPE Cells

Recently, our group showed that PM2.5 induces cellular stress by degenerating primary cilia in RPE cells and keratinocytes [17]. Thus, we further addressed the effects of 2-IPMA on ciliogenesis in PM2.5-treated RPE cells. RPE/Smo-GFP cells were treated with PM2.5 in the presence or absence of 2-IPMA and then primary ciliogenesis was observed. Consistent with previous reports, exposure to PM2.5 decreased both ciliated cells and cilium length in RPE cells (Figure 2). Importantly, the dysregulated primary cilia were markedly restored in combination with 2-IPMA in RPE cells, suggesting that 2-IPMA inhibits PM2.5-mediated ciliary dysgenesis. Since PM2.5 increases cellular damage via excessive generation of reactive oxygen species (ROS) in various tissues, we additionally investigated the effect of 2-IPMA on ROS production. Consistently, cellular ROS levels were considerably increased by treatment with PM2.5 in RPE cells. However, the excessive ROS were dramatically removed by treatment of RPE cells with either 2-IPMA or N-acetylcysteine (NAC), an ROS scavenger (Figure 3A,B). These results indicate that 2-IPMA has an antioxidant activity. We further examined this antioxidant activity during primary ciliogenesis in PM2.5-treated cells. As shown in Figure 3C,D, both 2-IPMA and NAC restored the PM2.5-induced loss of primary cilia, suggesting that the antioxidant activity of 2-IPMA recovered ciliary dysgenesis in PM2.5-treated cells.

### 3.3. 2-IPMA Inhibits Activation of JNK and Production of Proinflammatory Cytokines in PM2.5-Treated RPE Cells

We next addressed the potential regulatory mechanism of 2-IPMA-mediated ciliogenesis in PM2.5-treated cells. Since c-Jun NH2-terminal kinase (JNK), a stress response kinase, mediates PM2.5-induced cellular damage [17,22], we examined PM2.5-mediated JNK activation. As expected, PM2.5 treatment promoted JNK phosphorylation, which was blocked by SP600125, a JNK inhibitor (Figure 4A). In particular, treatment with 2-IPMA also suppressed JNK activation in PM2.5-treated cells as well as SP600125 (Figure 4A). Importantly, an immunostaining assay with primary cilia showed that the decrease in ciliated cells and cilium length was completely recovered by either SP600125 or 2-IPMA in PM2.5-treated cells (Figure 4B,C). In addition, we found that treatment of anisomycine, known as a JNK activator, suppressed the ciliogenesis induced by 2-IPMA in PM2.5-treated RPE cells (Figure 4D).

It has been reported that exposure to PM2.5 induces inflammation by activating various stress signals [23,24]. Moreover, the JNK pathway plays multiple roles in various inflammation-associated conditions [5]. Thus, we further investigated the role of 2-IPMA in the PM2.5-mediated inflammation response by measuring levels of proinflammatory cytokines, specifically IL-6 and TNF-α, in cell culture supernatants. Consistently, PM2.5 treatment increased expression levels of IL-6 and TNF- α, which were significantly suppressed by 2-IPMA in RPE cells (Figure 5A,B). Because IFT88 regulates anterograde IFT, loss of IFT88 disrupts cilia assembly. Thus, we additionally investigated the effects of primary ciliogenesis on inflammation in PM2.5-treated cells. Interestingly, we found that the reduction in proinflammatory cytokines by 2-IPMA was substantially restored by inhibition of primary ciliogenesis via *IFT88* knockdown in PM2.5-treated cells (Figure 5). Taken together, these results suggest that 2-IPMA recovers ciliary dysgenesis caused by PM2.5 by inhibiting JNK activation and inflammation in RPE cells.

The RPE cells are monolayered pigmented cells located just outside the neurosensory retina that nourish retinal visual cells; RPE is firmly attached to the underlying choroid and overlying retinal visual cells [25]. Therefore, dysfunction of the RPE is associated with various retinopathies including age-related macular degeneration and diabetic retinopathy [26]. Both oxidative stress and inflammation responses have been indicated as major toxic mechanisms underlying retinopathies and cancers [27,28]. Consistently, an injury caused by oxidative stress and inflammation on RPE cells can eventually cause retinal atrophy and subsequently lead to the pathogenesis of age-related macular degeneration [29,30,31]. Notably, PM2.5 can induce oxidative stress and inflammation in various epithelial cells [31,32,33]. It has been reported that rats exposed to PM2.5 show decreased retinal thickness [34]. Moreover, PM2.5 treatment causes retinal dysfunction by promoting the epithelial–mesenchymal transition of RPE cells [35], suggesting that PM2.5 induces retinal dysfunction.

In the current study, we identified 2-IPMA as a potent primary cilia inducer from a library screening. From this screening, we also identified known inducers of ciliogenesis, such as β-carotene, niacinamide, and N-acetylglucosamine, as well as 2-IPMA [36,37,38]. However, 2-IPMA is a metabolite that has yet to be studied in detail [39,40]. Previously, it was reported that 2-IPMA is an intermediate in the biosynthesis of leucine [21] and it has been identified in the metabolite profiles of white wines [39,40]. However, the physiological role played by 2-IPMA in cells had yet to be evaluated. Here, we showed that 2-IPMA has antioxidant and anti-inflammatory effects on PM2.5-induced cell stress by promoting primary ciliogenesis in RPE cells (Figure 3 and Figure 5).

Primary cilia regulate diverse cellular processes during signaling to maintain homeostasis [9]; thus, ciliary dysgenesis is a cause of various diseases, i.e., ciliopathies, including hydrocephalus, infertility, airway diseases, and polycystic diseases of the kidney, liver, and pancreas, as well as retinal diseases and defects of hearing and smell [41]. Recently, we reported that primary cilia reduce oxidative stress to promote cell survival in neuronal cells [42]. Blockade of primary ciliogenesis promotes neuronal loss and motor disability in Parkinson’s disease models [42]. Loss of primary cilia also promotes mitochondria-dependent cell death in thyroid cancer [12]. Consistently, Choi et al. showed that primary cilia enhance retinal ganglion cell survival after axotomy [43]. These results indicate that primary cilia have protective effects by reducing oxidative stress. In the present study, we found that 2-IPMA sufficiently removes excessive ROS production but almost completely restores ciliary dysgenesis induced by PM2.5 in RPE cells (Figure 3).

To evaluate the underlying mechanism of 2-IPMA in primary ciliogenesis, we investigated JNK, a cellular stress response kinase, and found that 2-IPMA strongly inhibited PM2.5-induced JNK activation (Figure 4). JNK contributes to various pathological conditions including neurodegenerative and metabolic diseases [44,45]. Although further studies are needed to elucidate the action mechanism, our group and other groups have recently demonstrated that the JNK pathway mediates oxidative stress in response to PM2.5 exposure [17,22]. Inhibition of JNK by a chemical inhibitor, SP600125, recovered PM2.5 exposure-induced dysregulated primary cilia in RPE cells [17]. Since JNK is involved in the action of 2-IPMA in PM2.5-treated cells, further investigations into the JNK activation mechanism related to signaling and molecular targeting of 2-IPMA will be required to elucidate the functions of 2-IPMA as an inducer of primary ciliogenesis.

It has been reported that the JNK pathway is also involved in PM2.5-triggered inflammatory responses in epithelial cells [46]. Numerous studies have demonstrated the significant induction of proinflammatory cytokines in diabetes patients; moreover, diabetic retinopathy is the most common microvascular complication of diabetes and the leading cause of retinopathy [47]. In addition, PM2.5 induces inflammation and cell death in human trabecular meshwork cells to trigger ocular hypertension and glaucoma [48]. Given these previous findings, we chose to assess proinflammatory cytokine levels and found that RPE cells exposed to PM2.5 showed a large increase in the secretion of IL-6 and TNF-α, which are both associated with retinopathy [27,47]. Moreover, 2-IPMA significantly ameliorated production of these inflammatory cytokines in PM2.5-treated RPE cells, which was reversed by inhibition of ciliogenesis (Figure 5). Our results indicate that 2-IPMA has protective effects against-PM2.5-induced inflammation, which arise by promotion of primary ciliogenesis in RPE cells.

In conclusion, we demonstrated that 2-IPMA ameliorates PM2.5-induced RPE dysfunction by promoting primary ciliogenesis. Although additional mechanistic and in vivo studies are still required, our findings suggest that 2-IPMA is a potential therapeutic agent for retinopathies associated with loss of primary cilia.

## Figures and Tables

**Figure 1 molecules-26-05409-f001:**
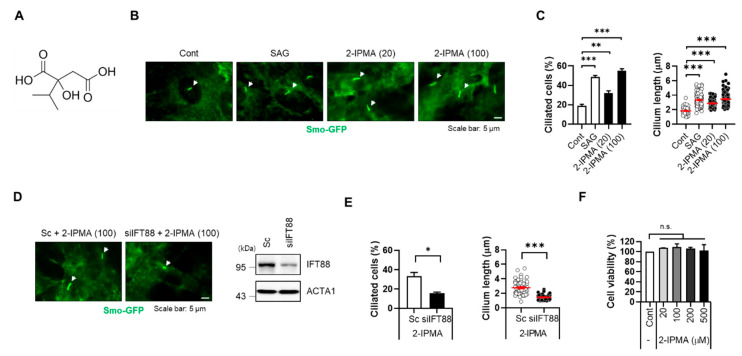
2-IPMA promotes primary ciliogenesis in RPE cells. (**A**) Chemical structure of 2-isoproylmalic acid (2-IPMA). (**B**,**C**) RPE cells stably expressing Smo-GFP (RPE/Smo-GFP) were treated with either SAG (1 µM) or 2-IPMA (20 or 100 µM) for 24 h. (**B**) Cells were imaged by fluorescence microscopy. White arrows indicate the primary cilia. (**C**) The ciliated cells and cilium length of the cells were measured under a fluorescence microscope. (**D**,**E**) RPE/Smo-GFP cells transiently transfected with either scrambled siRNA (Sc) or targeted siRNA for *IFT88* (si*IFT88*) were treated with 2-IPMA (100 µM) for 24 h. (**D**) Cells were imaged by fluorescence microscopy. (**E**) The ciliated cells and cilium length of the cells were measured. (**F**) RPE cells were treated with different concentrations of 2-IPMA and cell viability was measured by the CCK-8 assay after 24 h. Data are presented as the mean ± SEM (*n* = 3, * *p* < 0.05, ** *p* < 0.01, *** *p* < 0.001, n.s = not significant). Scale bar: 5 µm.

**Figure 2 molecules-26-05409-f002:**
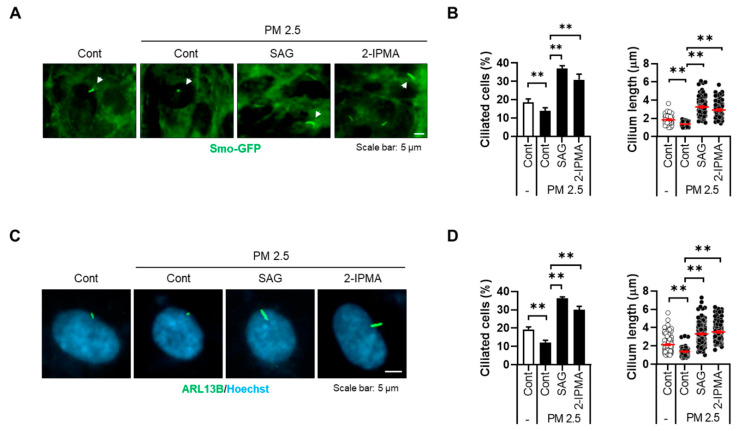
2-IPMA inhibits PM2.5-mediated ciliary dysgenesis in RPE cells. (**A**,**B**) RPE/Smo-GFP cells were exposed to PM2.5 (50 µg/mL) in the presence or absence of SAG (1 µM) and 2-IPMA (100 µM) for 24 h. (**A**) Cells were imaged by fluorescence microscopy. (**B**) The ciliated cells and cilium length of the cells were counted under a fluorescence microscope. (**C**,**D**) RPE cells were exposed to PM2.5 (50 µg/mL) in the presence or absence of SAG (1 µM) and 2-IPMA (100 µM) for 24 h. (**C**) The cells were stained with ARL13B antibody (green) and Hoechst dye (blue). (**D**) The ciliated cells and cilium length of the cells were measured. Data are presented as the mean ± SEM (*n* = 3, ** *p* < 0.01). Scale bar: 5 µm.

**Figure 3 molecules-26-05409-f003:**
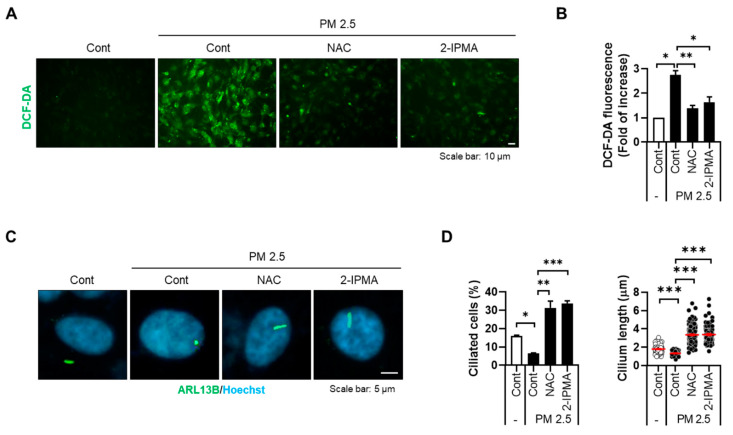
2-IPMA inhibits excessive ROS generation by PM2.5 in RPE cells. (**A**,**B**) RPE cells were treated with PM2.5 (50 µg/mL) in the presence of either 2-IPMA (100 µM) or NAC (10 µM) for 24 h. Then, the levels of cellular ROS were imaged (**A**) and measured with a DCFH-DA fluorescence ROS detection assay (**B**). Scale bar: 10 µm. (**C**,**D**) RPE cells were treated with PM2.5 (50 µg/mL) in the presence of either 2-IPMA (100 µM) or NAC (10 µM) for 24 h. (**C**) The cells were stained with ARL13B antibody (green) and Hoechst dye (blue). (**D**) The ciliated cells and cilium length of the cells were measured. Scale bar: 5 µm. Data are presented as the mean ± SEM (*n* = 3, * *p* < 0.05, ** *p* < 0.01, *** *p* < 0.001).

**Figure 4 molecules-26-05409-f004:**
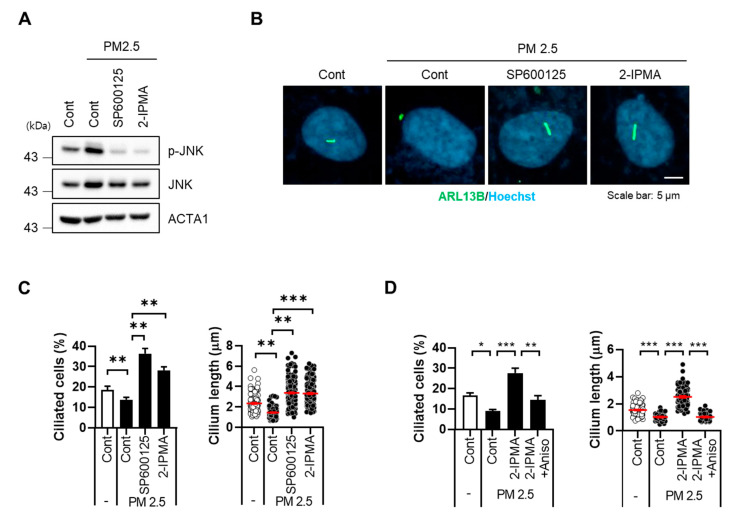
2-IPMA inhibits PM2.5-induced JNK activation in RPE cells. RPE cells were exposed to PM2.5 (50 µg/mL) with either 2-IPMA (100 µM)/SP600125 (10 µM) or 2-IPMA (100 µM)/anisomycin (1 µg/mL) for 24 h. Then the cells were harvested and analyzed by Western blotting with indicated antibodies (**A**). The cells were stained with ARL13B antibody (green) and Hoechst dye (blue) (**B**). Both ciliated cells and cilium length of the cells were measured under a fluorescence microscope (**C**,**D**). Data are presented as the mean ± SEM (*n* = 3, * *p* < 0.05, ** *p* < 0.01, *** *p* < 0.001). Scale bar: 5 µm.

**Figure 5 molecules-26-05409-f005:**
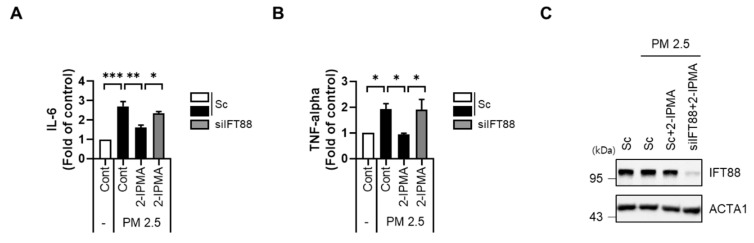
2-IPMA inhibits pro-inflammation cytokine production of IL-6 and TNF-α induced by PM2.5 in RPE cells. RPE cells transfected with either scrambled siRNA (Sc) or siRNA against *IFT88* (si*IFT88*) were further exposed to PM2.5 (50 µg/mL) with 2-IPMA (100 µM) for 24 h. Then the expression levels of IL-6 (**A**) and TNF-α (**B**) in the culture condition were measured using an ELISA assay. The reduced expression of IFT88 was measured by Western blotting (**C**). (*n* = 3, * *p* < 0.05, ** *p* < 0.01, and *** *p* < 0.001).

## Data Availability

Not applicable.

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
