# Peer review of "2-IPMA Ameliorates PM2.5-Induced Inflammation by Promoting Primary Ciliogenesis in RPE Cells"

_molecules, 2021, doi:10.3390/molecules26175409_

Round 1

Reviewer 1 Report

In this paper, Choi et al. found 2-IPMA as a novel ciliogenesis inducer. IPMA also blocks PM2.5-induced oxidative stress and inflammation response, which are associated with various pathological conditions. It is an interesting result and eligible to introduce in the ‘Molecules’. But some minor revisions will be helpful to improve the manuscript.

Minor comments:

1) The authors identified 2-IPMA from a screening with a metabolite library. What other molecules were identified from the screening as well as 2-IPMA?

2) In the paper, the authors used only RPE cells to examine the effect to 2-IPMA. Does 2-IPMA also potentially induces ciliogenesis in other types of cells?

3) Fluorescent protein and visualization probes facilitate biological monitoring. The cell-based screening fluorescent protein system established by the authors speeds up the identification of metabolites of ciliogenesis. Thus, it is more reasonable to state this sentence (Lines 158-160). "Visual detection tools for biological proteins and metabolites are essential (Chinese Chemical Letters, 2019, 30(10): 1704-1716.), to identify metabolites for regular ciliogenesis, we previously developed a cell-based screening system using RPE cells that stably express a fluorescent protein fused with Smo (RPE/Smo-GFP) [20]."

4) In Figures 1D and E, the authors need to present the knock-down efficiency of IFT88 siRNA.

5) In Figure 4, 2-IPMA phosphor-c-Jun activation by PM2.5. However, it was described as JNK in the figure and text. Correct it rightly.

6) In Figure 4. 2-IPMA inhibits JNK activation in PM2.5-treated cells. Reversely, does treatment of a JNK activator suppress ciliogenic effect of 2-IPMA in PM2.5-treated cells?

7) This sentence (Lines 191-192) should be slightly adjusted to “Since excessive reactive generation of reactive oxygen species (ROS) is notorious (PMID: 32981100), and is the reason why PM2.5 increases cellular damage in various organizations, we additionally investigated the effect of 2-IPMA on ROS production.”

Author Response

Response to Reviewers' comments: (Reviewer #1)

In this paper, Choi et al. found 2-IPMA as a novel ciliogenesis inducer. IPMA also blocks PM2.5-induced oxidative stress and inflammation response, which are associated with various pathological conditions. It is an interesting result and eligible to introduce in the ‘Molecules’. But some minor revisions will be helpful to improve the manuscript.

Minor comments:

Q1. The authors identified 2-IPMA from a screening with a metabolite library. What other molecules were identified from the screening as well as 2-IPMA?

Response 1: We appreciate your kind comment. From the screening, we also identified oo ooo oo as potent inducer for ciliogenesis. We described it in the Discussion part.

Q2. In the paper, the authors used only RPE cells to examine the effect to 2-IPMA. Does 2-IPMA also potentially induces ciliogenesis in other types of cells?

Response 2: We appreciate your kind comment. Not only RPE cell, 2-IPMA also efficiently induced ciliogenesis in neuroblastoma cells and fibroblasts.

Q3. Fluorescent protein and visualization probes facilitate biological monitoring. The cell-based screening fluorescent protein system established by the authors speeds up the identification of metabolites of ciliogenesis. Thus, it is more reasonable to state this sentence (Lines 158-160). "Visual detection tools for biological proteins and metabolites are essential (Chinese Chemical Letters, 2019, 30(10): 1704-1716.), to identify metabolites for regular ciliogenesis, we previously developed a cell-based screening system using RPE cells that stably express a fluorescent protein fused with Smo (RPE/Smo-GFP) [20]."

Response 3: We appreciate your generous comment. According to the referee’s suggestion, we correct the sentence in Result part.

Q4. In Figures 1D and E, the authors need to present the knock-down efficiency of IFT88 siRNA.

Response 4: According to the reviewer’s kind suggestion, we confirmed the knockdown of IFT88 by Western blotting and added in Figure 1D.

Q5. In Figure 4, 2-IPMA phosphor-c-Jun activation by PM2.5. However, it was described as JNK in the figure and text. Correct it rightly.

Response 5: We appreciate your kind comment. There was a typo-error. We corrected it to JNK.

Q6. In Figure 4. 2-IPMA inhibits JNK activation in PM2.5-treated cells. Reversely, does treatment of a JNK activator suppress ciliogenic effect of 2-IPMA in PM2.5-treated cells?

Response 6: We appreciate to your valuable comment, As shown in revised Figure 4D, treatment of anosomycine, a chemical activator for JNK further inhibited the ciliogenesis induced by 2-IPMA in PM2.5-treated RPE cells.

Q7. This sentence (Lines 191-192) should be slightly adjusted to “Since excessive reactive generation of reactive oxygen species (ROS) is notorious (PMID: 32981100), and is the reason why PM2.5 increases cellular damage in various organizations, we additionally investigated the effect of 2-IPMA on ROS production.”

Response 8: We appreciate your generous comment. According to the referee’s suggestion, we correct the sentence.

Reviewer 2 Report

In this study, the authors identified  the 2-isopropylmalic acid (2-IPMA) as a novel inducer of primary ciliogenesis, demonstrating that it restores the PM2.5-induced dysgenesis of primary cilia in RPE cells. Starting from previous evidences, showing that PM2.5 promotes oxidative stress and inflammation via JNK pathway activation, the authors showed that in cells -treated with PM2.5,  2-IPMA exposure recovered ciliary dysgenesis, by inhibiting proinflammatory cytokines and ROS production, via JNK inhibition.

Collectively, the results suggest that  in RPE cells, PM2.5 exposure promotes inflammation and oxidative stress  via JNK activation, leading to ciliary injury and that 2-IPMA inhibited JNK activation, recovering ciliary dysgenesis.

The study is interesting. The authors used appropriate methodological approaches to demonstrate the aim of the study.

Minor revision:

  • The authors should better elucidate why they used 50µg of PM2.5
  • In the figure 4A the authors should use “p-JNK and Total JNK”, not p-c-Jun and c-Jun.

Author Response

Response to Reviewers' comments: (Reviewer #2)

In this study, the authors identified the 2-isopropylmalic acid (2-IPMA) as a novel inducer of primary ciliogenesis, demonstrating that it restores the PM2.5-induced dysgenesis of primary cilia in RPE cells. Starting from previous evidences, showing that PM2.5 promotes oxidative stress and inflammation via JNK pathway activation, the authors showed that in cells -treated with PM2.5, 2-IPMA exposure recovered ciliary dysgenesis, by inhibiting proinflammatory cytokines and ROS production, via JNK inhibition. Collectively, the results suggest that in RPE cells, PM2.5 exposure promotes inflammation and oxidative stress via JNK activation, leading to ciliary injury and that 2-IPMA inhibited JNK activation, recovering ciliary dysgenesis.

The study is interesting. The authors used appropriate methodological approaches to demonstrate the aim of the study.

Minor revision:

Q1. The authors should better elucidate why they used 50 µg of PM2.5

Response 1: We appreciate your valuable comments on our manuscript. Previous our report and other literatures showed that 50 µg of PM2.5 slightly induced cell death with oxidative stress in RPE cells (Bae JE et al., 2019; Lee H et al., 2020). According to this notion, we also treated PM2.5 (50 µg/ml) in RPE cells in this study.

Q2. In the figure 4A the authors should use “p-JNK and Total JNK”, not p-c-Jun and c-Jun.

Response 2: We appreciate your kind comment, according to the reviewer’s mentions, we corrected it to JNK instead of c-Jun.

Reference:

  1. Bae JE et al., Fine particulate matter (PM2.5) inhibits ciliogenesis by increasing SPRR3 expression via c-Jun activation in RPE cells and skin keratinocytes Sci Rep. 2019 Mar 8;9(1):3994.
  2. Lee H et al., Diesel particulate matter2.5 promotes epithelial-mesenchymal transition of human retinal pigment epithelial cells via generation of reactive oxygen species. Environ Pollut. 2020 Jul;262:114301.